# Preparation of Molecularly Imprinted Polymer Microspheres for Selective Solid-Phase Extraction of Capecitabine in Urine Samples

**DOI:** 10.3390/polym14193968

**Published:** 2022-09-22

**Authors:** Renyuan Song, Jiawei Xie, Xiaofeng Yu, Jinlong Ge, Muxin Liu, Liping Guo

**Affiliations:** 1School of Materials and Chemical Engineering, Bengbu University, Bengbu 233030, China; 2Anhui Provincial Engineering Laboratory of Silicon-Based Materials, Bengbu 233030, China; 3Functional Powder Material Laboratory of Bengbu City, Bengbu 233030, China

**Keywords:** molecularly imprinted solid-phase extraction, interaction forces, suspension-iniferter polymerization, capecitabine

## Abstract

Molecularly imprinted solid-phase extraction to treat biological samples has attracted considerable attention. Herein, molecularly imprinted polymer (MIP) microspheres with porous structures were prepared by a combined suspension-iniferter polymerization method using capecitabine (CAP) as a template molecule. This material was subsequently used as a solid-phase extraction agent to separate and enrich drug molecules in urine samples. UV analysis revealed that methacrylate (MAA) was an ideal functional monomer, and 1H Nuclear Magnetic Resonance (^1^H NMR), Ultraviolet (UV), and Fourier transform-infrared (FT-IR) spectroscopic analyses were used to study the interaction forces between MAA and CAP, demonstrating that hydrogen bonding was the primary interaction force. MIPs with outstanding selectivity were successfully prepared, and the analysis of their surface morphology and chemical structure revealed a spherical morphology with small holes distributed across a rough surface. This surface morphology significantly reduced the mass transfer resistance of template molecules, providing an ideal template recognition effect. Using the molecularly imprinted solid-phase extraction method, CAP and the structural analog cytidine (CYT) were pretreated in urine samples and quantified by HPLC. The results showed that CAP and CYT recoveries reached 97.2% and 39.8%, respectively, with a limit of detection of 10.0–50.0 µg·mL^−1^. This study provides a novel approach to drug molecule pretreatment that can be applied in drug separation and functional materials science fields.

## 1. Introduction

Capecitabine (CAP) is an oral 5-fluorouracil (5-FU) prodrug that is commonly used to treat patients with colorectal and breast cancer [1,2]. The entire molecule can be rapidly and completely absorbed through the gastrointestinal wall after ingestion; therefore, the amount of residual drug in bodily fluids can be used to evaluate its pharmacokinetics [3,4]. At present, direct detection is not typically feasible due to sample complexity and low drug concentration [5]. Therefore, appropriate sample-preparation techniques are required for an improved detection and quantification [6]. Common sample pretreatment methods include conventional liquid-liquid extraction (LLE) and solid-phase extraction (SPE), which are often tedious, time-consuming, and require large amounts of toxic organic solvents [7,8]. Therefore, the rapid, efficient, and sustainable development of novel sample pretreatment technologies has important practical implications in the field of drug testing.

The specific affinity of molecularly imprinted polymers (MIPs) enables the selective adsorption of target molecules and their structural analogs from complex samples. MIPs can be used as solid-phase extractants to selectively separate, extract, and enrich trace drug molecules from complex samples [9,10,11,12]. Trace analyte detection must overcome unfavorable factors, such as complex medical, biological, and environmental sample matrices and cumbersome pretreatment protocols. Thus, complex sample pretreatment and enrichment processes are often employed. Recently, the integration of MIPs and SPE technology has become increasingly popular [13,14,15,16]. Utilizing MIP materials as a special SPE adsorption agent can further reduce endogenous interference caused by an interferent in the sample by enriching the analyte and successfully eliminating matrix interference [17]. The purified samples can be directly injected into GC, HPLC, GC-MS, HPLC-MS, and other analytical instruments. Molecularly imprinted solid-phase extraction (MISPE) has attracted extensive attention because of its straightforward operation, rapid reaction, high enrichment factor, and facile automation potential [18,19,20,21].

MISPE technology is proposed herein to separate capecitabine (CAP) and its structural analogs in urine samples to examine the method’s suitability for drug testing in biological samples. Herein, we propose the use of template molecules to prepare porous MIP microspheres using combined suspension-iniferter polymerization, which can be applied as an adsorption material for SPE for the enrichment and separation of CAP in urine. Then, ^1^H NMR, UV, and FT-IR spectroscopic methods were used to evaluate the interaction forces between the template molecule and functional monomer to select a suitable functional monomer for the selective MIP material preparation. To further analyze the specific recognition performance of the prepared MIPs, cytidine (CYT) was selected as an interferent for CAP in MISPE-pretreated urine samples. The sensitivity and selectivity of this newly developed method were also assessed.

## 2. Materials and Methods

### 2.1. Materials

CAP, CYT, acrylamide (AM), 4-vinyl pyridine (4-VP), and methacrylic acid (MAA) were purchased from Shanghai Aladdin Reagent Co., Ltd. (Shanghai, China). Sodium decylalkyl sulfate (SDS) and cetanol (CA) were obtained from Tianjin Kemiou Chemical Reagent Co., Ltd. Yantai Yunkai Chemical Co., Ltd. (Yantai, China). supplied ethylene glycol dimethacrylate (EGDMA). All other reagents were obtained from Sinopharm Chemical Reagent Co., Ltd. (Shanghai, China). All reagents were analytically pure. *N,N*-diethyl dithiocarbamate (BDC) was synthesized following a previously reported procedure [22].

### 2.2. Characterization

A Shimadzu UV-3900 spectrophotometer (Shimadzu, Japan,), Nicolet IS 10 FT-IR (Thermo Fisher Scientific, Waltham, MA, USA), and AVANCE NEO 400 M ^1^H NMR (Bruker, Switzerland) were used to examine interaction forces between the template molecules and functional monomers. The obtained polymer chemical structures were identified using a Nicolet IS 10 FT-IR spectrometer via the KBr compressed pellet method. The morphologies and structures of the polymer samples were determined using SEM (Zeiss Merlin Compact, Germany). Nitrogen adsorption tests were performed using an ASAP 2020 surface area and porosity analyzer (Micromeritics, Atlanta, GA, USA). Finally, biological sample analysis was performed using a Shimadzu Prominence LC-20A HPLC system (Shimadzu, Japan) equipped with a UV detector. 

### 2.3. UV Determination

DMSO solvent was used to prepare a 0.25 mmol^−1^ CAP solution, and the same concentration of functional monomer (MAA, AM, and 4-VP) solution was prepared. The solutions were mixed and shaken for 4 h at a volume ratio of 1:1, allowing for a full interaction that was monitored over the wavelength range of 190–400 nm. After screening for the appropriate functional monomer, mixed solutions of CAP and the selected functional monomer at ratios ranging from 1:0 to 1:8 were prepared and mixed completely and were left for a period of time. The corresponding concentrations of select functional monomer were monitored in the wavelength range of 200–400 nm and examined to distinguish the changes in the CAP UV spectrum.

### 2.4. Synthesis of MIP Microspheres

MIP microspheres were prepared using a combined suspension-iniferter polymerization method. The procedure was performed as follows (Figure 1): PVA (0.5 g) in deionized water (100 mL) was heated to 90 °C and stirred until it was completely dissolved to obtain a homogeneous aqueous solution. A mixture of CAP (0.7187 g), DMSO (20 mL), and MAA was placed in a 250 mL four-neck round-bottom quartz flask equipped with an argon inlet tube and mechanical stirrer. After 3 h, the mixture was pre-polymerized to obtain a clear, uniform mixture. An appropriate amount of cross-linker EGDMA and initiator BDC was added, and the prepared aqueous solution was added after ultrasonic mixing. The reaction was maintained for 1 h to obtain uniform small droplets. The flask was then placed in a microwave-ultraviolet-ultrasonic trinity synthesis extraction apparatus (MUUTSEA, UWave-1000, Shanghai, China), and the polymerization reaction was performed under microwave heating (180 W) and mechanical stirring at 400 rpm for 10 h. Upon reaction completion, the obtained polymers were washed several times with hot water (>90 °C) to remove unreacted monomer and free stabilizer. Subsequently, a methanol/acetic acid (9/1, *v*/*v*) mixture was used to remove the template molecules until CAP residues were no longer detected. Finally, the obtained MIP microspheres were dried under vacuum at 30 °C for 48 h.

Corresponding non-imprinted polymer (NIP) microspheres were prepared under the polymerization conditions described above, except that the template molecule was not added.

### 2.5. Preparation of MISPE

Dried MIPs/NIPs (500 mg) were loaded onto a 6 mL SPE column (the extraction column material was polypropylene). The extraction column ends were sealed using a porous polytetrafluoroethylene sieve plate. Before use, the MISPE column was eluted with a mixture of methanol-acetic acid (7/3, *v*/*v*) at a flow rate of 0.5 mL·min^−1^, and the eluent was collected until the template molecule could not be detected by HPLC. It was then further washed with excess methanol and dried under vacuum at 50 °C. All the columns were preconditioned with acetonitrile (3 mL) and deionized water (3 mL) in sequence.

### 2.6. Pretreatment of Urine Samples

After centrifugation at 5000 rpm for 10 min, the supernatant was collected and stored in a brown EP tube and frozen at −20 °C until further use. Then, 10 mg MIPs/NIPs were weighed and placed in a 50 mL conical flask with a methanol solution (5 mL) of mixed substrate (0.2 mg∙mL^−1^ for both CAP and CYT) (Figure 2). Subsequently, 10 mL of frozen urine was added to the conical flask described above, shaken and adsorbed for 10 h. After high-speed centrifugation, the supernatant was taken as a urine sample for testing.

The treatment method of the selected urine samples was as follows: 50 μL of the urine sample was added to a 1.5 mL brown EP tube with 1% aqueous acetic acid (15 μL) and acetonitrile (100 μL). After eddy current processing for 1 min, the samples were centrifuged at 4000 rpm for 10 min, 50 μL of the supernatant was collected, and 10 μL of the supernatant was injected into the HPLC.

## 3. Results and Discussion

### 3.1. Interaction between the Template Molecule and Functional Monomer

MIPs can be classified as covalent or non-covalent, divided based on the type of interactions between the functional monomers and template molecules that occur to form host-guest complexes [23]. MIPs prepared via covalent bonding have the disadvantage of a difficult template molecule elution and poor specific recognition. Thus, the number of MIPs prepared by this method is limited [24]. In contrast, non-covalent MIPs facilitate template elution and can rapidly recognize template molecules, achieving high affinities and distinct selectivities. Consequently, the degree of binding between the template molecules and functional monomers before polymerization is an important factor [25,26]. The non-covalent forces mainly involve hydrogen bonds, metal coordination bonds [27], ionic bonds [28], hydrophobic interactions [29], π-π interactions, and van der Waals forces [30]. These forces are essential for forming stable host-guest complexes via self-assembly.

In the imprinted polymerization system studied herein, the template molecule CAP contains numerous functional groups, including -OH, C=O, NH-, and other active groups in its structure that strongly interact with the functional monomer active groups. To obtain highly selective non-covalent MIPs and determine interaction forces to better understand template recognition mechanisms, UV, FT-IR, and ^1^H-NMR analysis methods were applied [31,32].

In the pre-polymerization system, the interaction forces between the functional monomers and CAP template were analyzed using UV spectrophotometry. In the spectrum depicted in Figure 3a, a characteristic UV absorption peak attributed to the simple geometrical addition of CAP and 4-VP appeared at 214 nm, whereas the characteristic peak of the CAP and 4-VP complex was redshifted by 3.5 nm to 217.5 nm. This indicates a strong interaction force between CAP and 4-VP. As illustrated in Figure 3b, the characteristic UV absorption peak attributed to the simple geometric addition of CAP and AM appeared at 214 nm, while the characteristic peak of the CAP and AM host-guest complex appeared at 213 nm. A slight blue shift was observed, indicating weak interaction forces between CAP and AM, preventing a stable host-guest complex formation. As shown in Figure 3c, the characteristic UV absorption peak of the CAP and MAA host-guest complex appeared at 210 nm, with a significant redshift of 6.5 nm. A comparison of the interaction forces between the three monomers (4-VP, AM, and MAA) and CAP revealed that the force between MAA and CAP was the strongest, forming a stable CAP-MAA host and guest complex. Therefore, MAA was selected as the functional monomer to prepare the MIPs.

UV spectroscopy is often used to study the interaction forces between template molecules and functional monomers [33]. To further examine the interaction force between CAP and MAA, a UV spectroscopy analysis on a mixed solution of CAP and MAA was performed at different ratios. As shown in Figure 4, from 200 to 350 nm, CAP exhibited three absorption peaks at 204, 243, and 309 nm. The two higher wavelength characteristic absorption peaks arise from the six-membered rings in the molecular structure. The interaction force between the six-membered rings and MAA is weak, and MAA addition did not change the position of these distinct absorption peaks. Nevertheless, when different ratios of MAA were added, the characteristic absorption peak at 204 nm shifted significantly from 204 to 214 nm, indicating a strong interaction between CAP and MAA. The absorption peaks of C=O and NH- groups at 204 nm mainly originate from the molecular structure of CAP. When MAA is added, the -OH or C=O groups in its structure easily form hydrogen bonds with the active groups in CAP.

To confirm the interaction between template CAP and MAA, FT-IR was used to study the CAP-MAA host-guest complex. As shown in Figure 5a, the peaks at 3530, 1710, and 1635 cm^−1^ correspond to the stretching vibration peaks of the carboxy hydroxyl group, carbonyl group, and C=C double bond, respectively. Simultaneously, the C-H out-of-plane bending vibration peak of the double bond in the MAA molecule appeared at 790 cm^−1^. As shown in Figure 5b, the peaks at 3500, 1750, 1650, and 1313 cm^−1^ arise from the expansion vibration of the hydroxyl group in CAP, base, amide bond, and C-F expansion vibration, respectively. As shown in Figure 5c, when the template molecule CAP was added at the same concentration, the C=O stretching vibration peaks of CAP and MAA shifted to 1702 cm^−1^, while the C-O stretching vibration peaks of CAP moved from 1204 to 1184 cm^−1^. A new absorption peak also appeared at 2555 cm^−1^, indicating that the hydroxyl groups in CAP and MAA underwent a novel hydrogen-bond formation. The above evidence confirms that CAP and MAA form hydrogen-bonding interactions.

^1^H NMR has been used to further examine the hydrogen-bonding forces between template molecules and functional monomers [34]. Under the same assay conditions, two absorption peaks appeared in the ^1^H-NMR spectrum of CAP at a chemical shift of 11.69, while a sharp singlet was exhibited at 10.51, as shown in Figure 6a. These peaks mainly arose from the two active hydrogens that were differently affected by the surrounding structure. The sharp absorption peak of the secondary hydrogen in CAP was observed at the chemical shifts of 5.05 and 5.44. The active hydrogen at this position is subjected to an isolated lone pair and base p-electron on N conjugation, so that the C-N bond exhibits a partial double-bond character, resulting in a sharp single peak. After adding MAA, two active hydrogens on the hydroxyl group in CAP (δ = 11.69, 10.51) and MAA active hydrogen (δ = 12.38) both occurred at a chemical shift of 12.18, as shown in Figure 6c. This is likely because the hydrogen bonds are chemically identical. The secondary hydrogen (δ = 5.04) in CAP exhibits two broad adsorption peaks, while the hydroxyl group forms a hydrogen bond with the carboxyl group in MAA.

### 3.2. FT-IR Analysis of MIP Microspheres

Figure 7 shows a broad absorption peak in the FT-IR spectra of MIPs and NIPs from 3300 to 3700 cm^−1^, with the stretching vibration peak arising from the carboxy hydroxyl group in MAA from 3500 to 3700 cm^−1^ and the hydroxyl group absorption from the CAP molecule in the IR MIP spectrum. The N-H stretching vibration peak of the secondary amine in CAP was also observed at 3200 cm^−1^. Thus, the superimposition of the amino, hydroxyl, and hydroxyl peaks in MAA occurs at 3200–3700 cm^−1^ [35]. The 2950 cm^−1^ absorption peak was observed after carboxy hydroxyl association. The stretching vibration absorption peak corresponding to C=O in the cross-linker agent EGDMA appeared at 1727 cm^−1^, while the C=C bond stretching vibration peak appeared at 1636 cm^−1^. The stretching vibration peaks of -COOH and C=S in the iniferter appeared at 1453 and 1163 cm^−1^, respectively [36]. This indicated that the CAP molecularly imprinted polymer was successfully prepared by combined suspension-iniferter polymerization.

### 3.3. SEM Analysis of the MIP Microspheres

The surface morphologies of the MIP and NIP microspheres were analyzed by SEM, and the results are shown in Figure 8. The SEM images show that the MIPs exhibited a good regularity and excellent monodispersity. Moreover, a significant porosity was observed on the MIP surfaces, which were rough with a loose structure that was imparted during the preparation process. The template molecule plays a significant role in pore formation, since after it is eluted a channel with a porous structure is formed, which can facilitate the mass transfer of the target. It is evident from the SEM image of NIPs that the surface was smoother and that the surface pores were less distributed than those of the MIPs. The particle size distribution was also more uniform, implying that the template molecule addition to the imprinting system affected the average particle size and porous structure formation on the surface of the polymer.

### 3.4. N_2_ Adsorption Analysis of the MIP Microspheres

As shown in Table 1, the specific surface area of MIPs (117.53 m^2^·g^−1^) was determined to be significantly larger than that of NIPs (69.85 m^2^·g^−1^), likely due to template addition. The imprinted polymerization system increased the polymer chain growth rate and affected the viscosity. The numerous template molecules present during MIP curing occupied positions in the polymer, becoming cavities after the elution and thus increasing the number of pores on the surface. The MIPs exhibited a high specific surface area and adsorption space, demonstrating a strong adoption ability. Moreover, the average pore size was 25.25 nm, which is considerably larger than the size of the CAP template molecule. CAP can freely enter and exit the pore channel with a low steric hindrance, which accelerates the molecular recognition rate and imparts an extremely high molecular recognition ability.

As shown in Figure 9, a hysteresis loop formed between the N_2_ adsorption and desorption curves of the MIPs, indicating a uniform porous structure on their surfaces. At low pressure, the adsorption capacity increased slowly, with N_2_ molecules adsorbed in the monolayer to form a multilayer on the inner surface of the MIPs channels. When the relative pressure P/P_0_ was >0.85, the gas adsorption capacity increased linearly, indicating that large pores were formed in the MIPs, which could significantly affect the MIPs along with the pore structure. The adsorption-desorption curve of NIPs showed a sharp increase when P/P_0_ was >0.8, with a clear hysteresis ring likely caused by the poor homogeneity of the pore structure in the polymer, which contained more large pores or more of an accumulation of particles.

### 3.5. MISPE Analysis of CAP in Urine Samples

As shown in Figure 10 and Figure 11, the impurities observed in the HPLC in the urine and standard samples after CAP-MISPE column treatment were lower, with CAP and CYT recovery rates of 97.2% and 39.8%, respectively. The recovery rate of CYT was low, while that of CAP was very high, implying that the CAP-MIPs prepared by the combined suspension-iniferter polymerization method exhibited outstanding specific recognition effects for template molecules. In addition, the recognition effect for structural analogs was low, indicating that MIPs could specifically recognize target molecules in complex environments, which is ideal for real-world applications. The detection limits of this method ranged from 10.0 to 50.0 µg·mL^−1^. The recovery rates of CPA and CYT by NIPs was poor because of the absence of imprinted holes and recognition sites matching the target molecules, with only an undifferentiated non-specific adsorption. These results demonstrate the potential applicability of MIPs for the efficient concentration, separation, and accurate quantification of CAP in complex biological samples.

## 4. Conclusions

This study proposed a novel general method for preparing MISPE using porous MIP microspheres. MIP microspheres were prepared using a combined suspension-iniferter polymerization method. The interaction between different functional monomers and CAP was investigated using UV spectroscopy, and the best functional monomer was determined to be MAA. Furthermore, the interaction forces between MAA and CAP at different ratios were analyzed by UV, ^1^H NMR, and FT-IR spectroscopic techniques; the mechanisms of interaction between the template molecules and functional monomers were elucidated, and the optimal mixing ratio was determined simultaneously. The surface morphology and chemical structure of the MIP materials were characterized by SEM, FT-IR spectroscopy, and N_2_ adsorption-desorption experiments. The MIPs exhibited regular spherical structures with a rough surface and many evenly distributed small pores, significantly reducing the number of template molecules. The mass transfer resistance of the template molecules was greatly reduced, yielding an ideal recognition effect with MIPs, whereas the NIP surfaces were smoother with fewer small pores, which was mainly caused by the absence of molecular imprinting. Finally, MIP and NIP microspheres were used as SPE materials for the separation and enrichment of drug molecules in urine samples. A subsequent HPLC analysis demonstrated a high recovery and low detection limit for the representative drugs. Thus, MISPE is a promising affinity material for the specific analysis of drugs in complex biological samples.

## Figures and Tables

**Figure 1 polymers-14-03968-f001:**
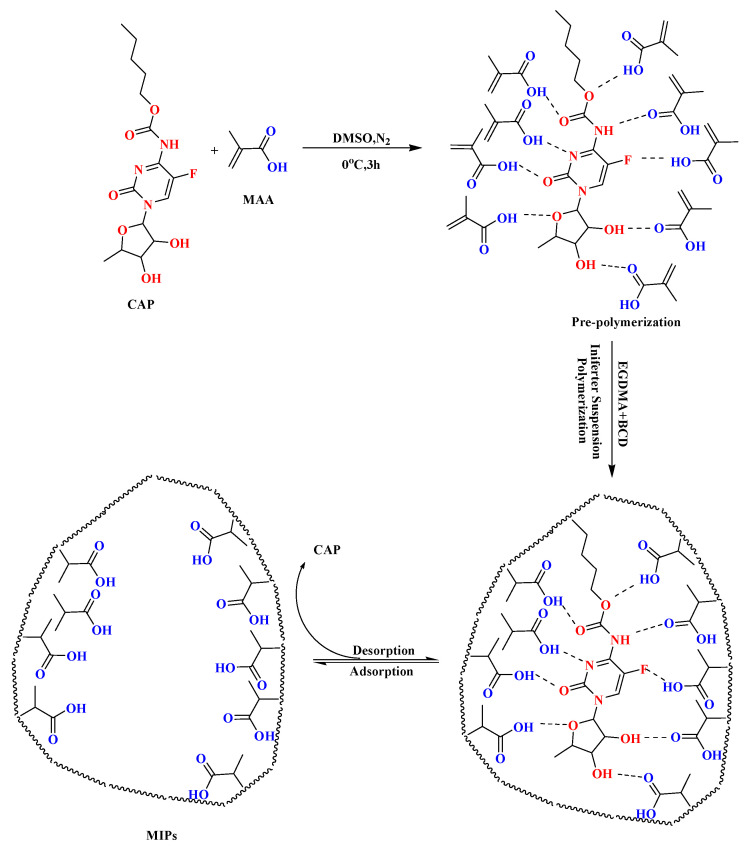
CPA-imprinted polymer microspheres were synthesized via a combined suspension-iniferter polymerization method.

**Figure 2 polymers-14-03968-f002:**
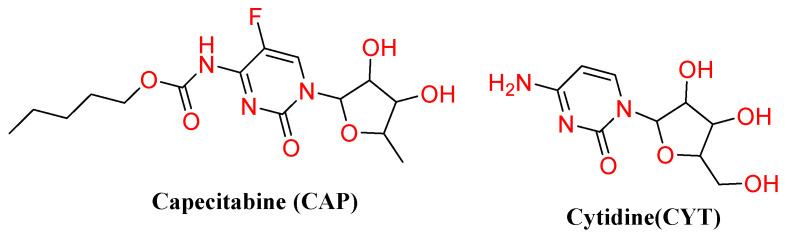
Chemical structures of CAP and CYT.

**Figure 3 polymers-14-03968-f003:**
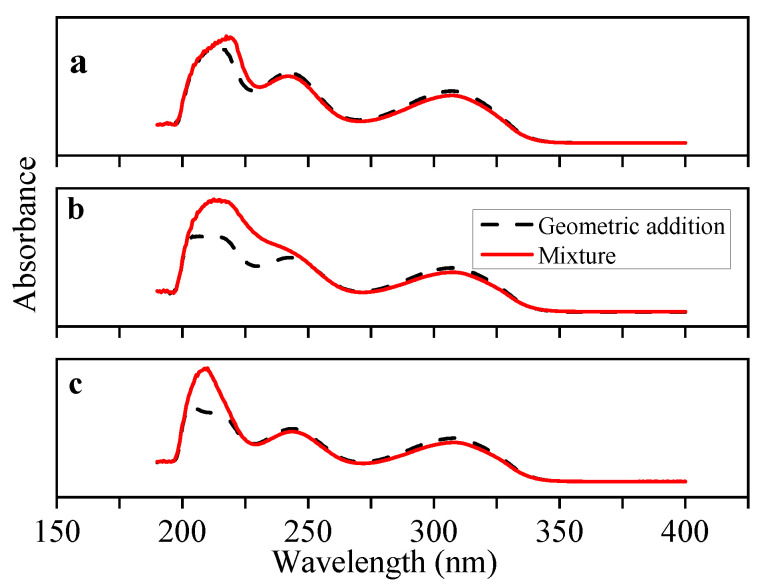
UV absorption spectra of CAP and host-guest coordination compound ((**a**) CAP-4-VP, (**b**) CAP-AM, (**c**) CAP-MAA) in DMSO.

**Figure 4 polymers-14-03968-f004:**
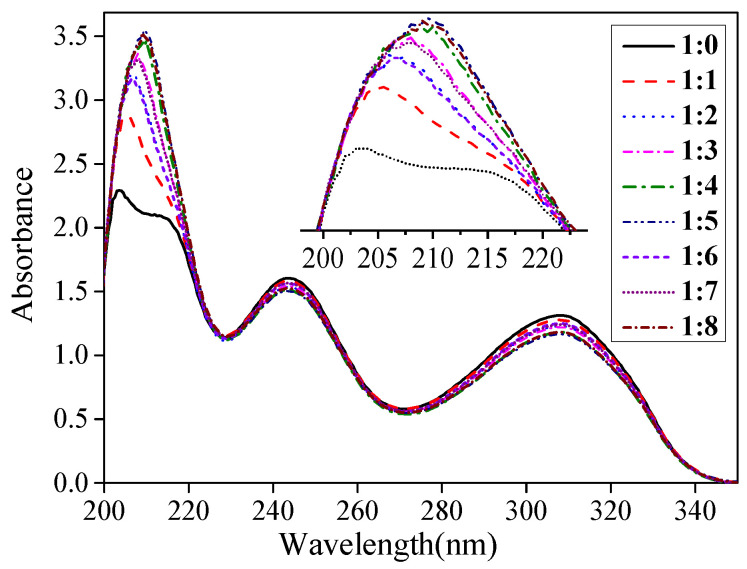
UV absorption spectra of CAP and MAA mixed at different molar ratios in DMSO.

**Figure 5 polymers-14-03968-f005:**
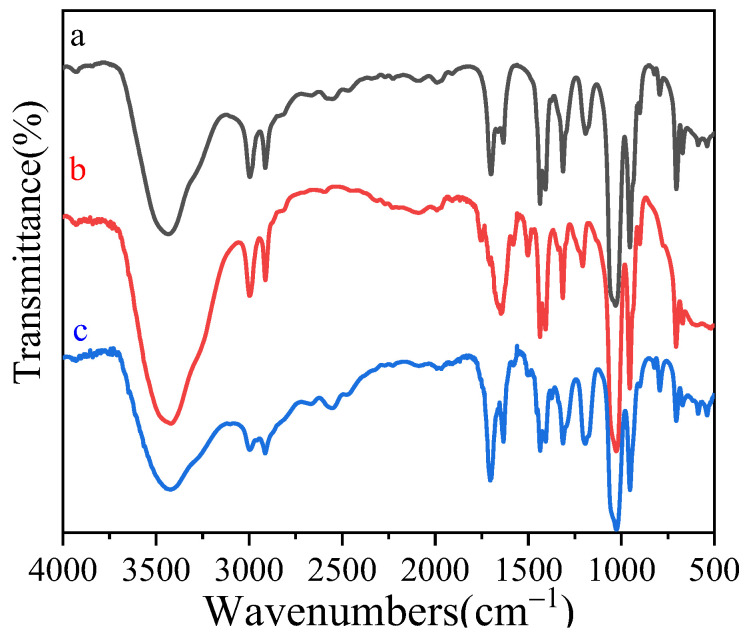
FT−IR spectra of (a) MAA, (b) CAP, and (c) CAP−MAA.

**Figure 6 polymers-14-03968-f006:**
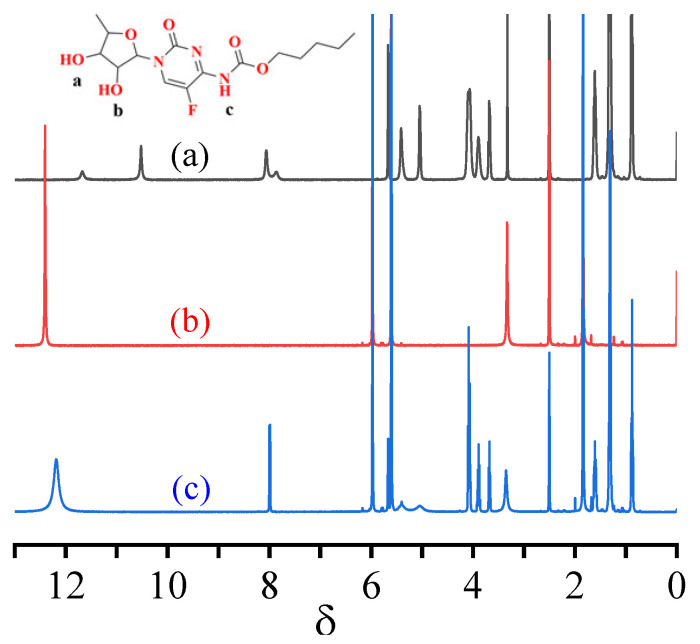
H-NMR spectra of CAP (a), MAA (b), and CAP-MAA (c) in d_6_-DMSO.

**Figure 7 polymers-14-03968-f007:**
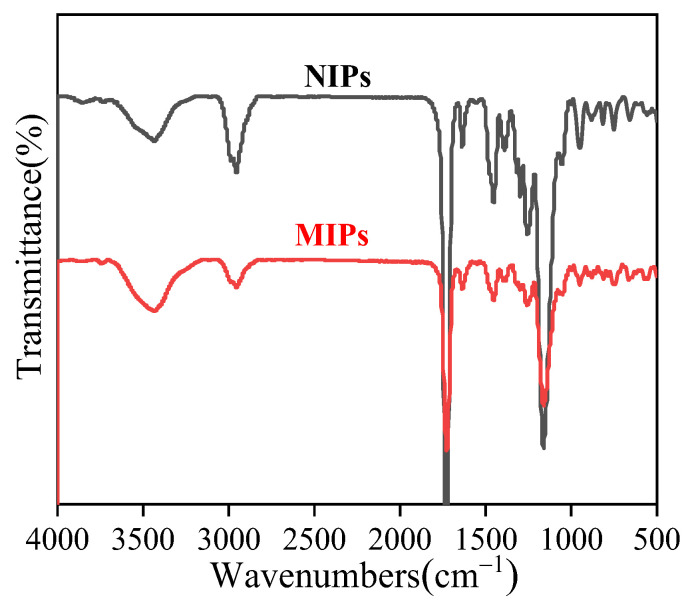
FT−IR spectra of the MIPs and NIPs microspheres.

**Figure 8 polymers-14-03968-f008:**
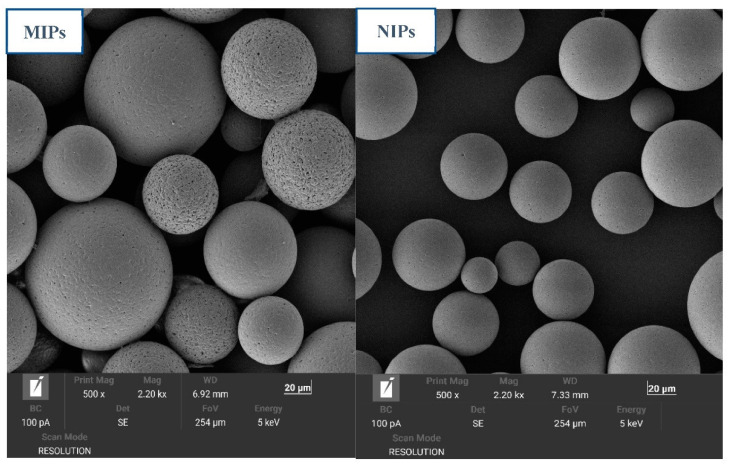
SEM images of the MIP (**left**) and NIP (**right**) microspheres.

**Figure 9 polymers-14-03968-f009:**
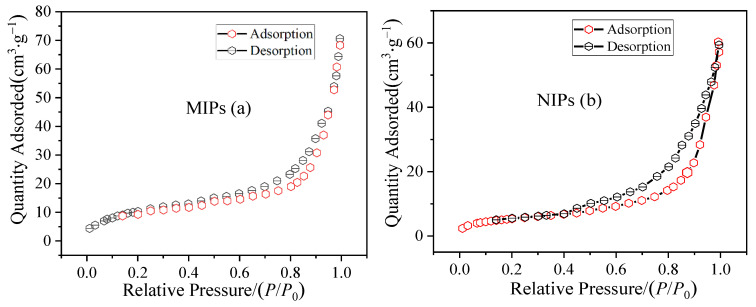
N_2_ adsorption−desorption curves of (**a**) MIPs and (**b**) NIPs.

**Figure 10 polymers-14-03968-f010:**
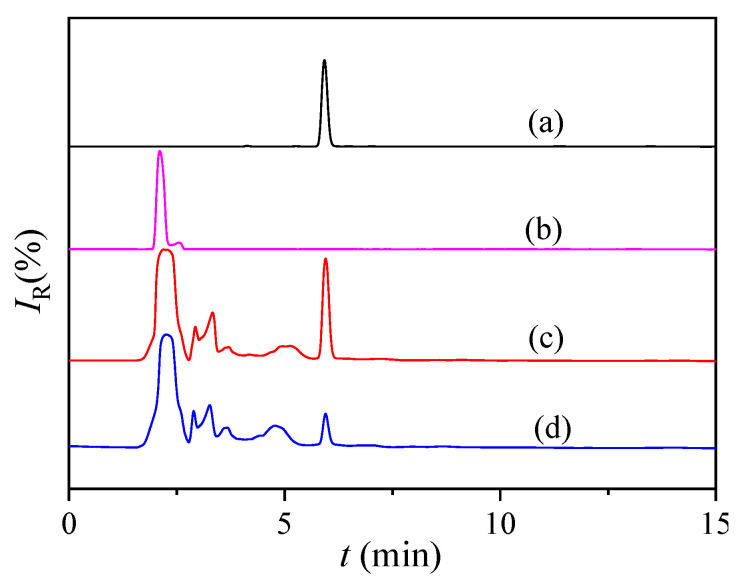
The chromatogram obtained after MIPs/NIPs-SPE pre-concentration of urine samples spiked with 10 μg∙mL^−1^ of CAP and CYT, respectively. (a) CAP standard solution, (b) CYT standard solution, (c) solution after MIPs-SPE treatment and (d) solution after NIPs-SPE treatment. Chromatographic conditions include C18 reversed-phase column: mobile phase, acetonitrile-H_2_O (35/65, *v*/*v*); flow rate, 0.5 mL∙min^−1^; UV detector wavelength, 310 nm; column temperature, 30 °C.

**Figure 11 polymers-14-03968-f011:**
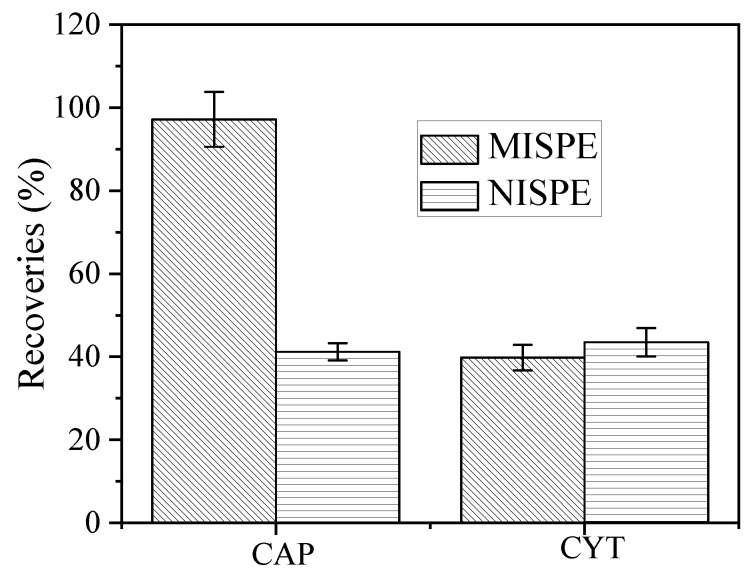
Recoveries of CAP and CYT on MISPE and NISPE cartridges.

**Table 1 polymers-14-03968-t001:** Surface areas, pore volumes, and pore sizes of the MIP and NIP microspheres.

Adsorbent	*d*_p_ (nm)	*V*_p_ (cm^3^·g^−1^)	*S* (m^2^·g^−1^)
NIPs	13.17	0.0462	69.85
MIPs	25.25	0.0846	117.53

## Data Availability

The data presented in this study are available on request from the corresponding author.

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
