# Peer review of "Preparation of Molecularly Imprinted Polymer Microspheres for Selective Solid-Phase Extraction of Capecitabine in Urine Samples"

_polymers, 2022, doi:10.3390/polym14193968_

Round 1

Reviewer 1 Report

In the present manuscript, molecularly imprinted polymer-based microspheres were synthetized and applied as SPE sorbents for capecitabine analysis in urine samples.

Some following issues should be carefully considered to further improve the quality of this paper:

·         The first paragraph of the introduction should be rephrased and discussed about the challenges which occur in analyzing biological samples and not food samples.

·         “MISPE technology is proposed to separate capecitabine (CAP) and its structural analogs in urine samples” – in this study the structural analogues of CAP are not enriched using the developed MIP, only the template molecules. There is no difference between the MIP and NIP behavior towards CYT.

·         There are some typographical mistakes throughout the manuscript: “polymerizatio,” – in the last paragraph of the introduction, “39.8%, respectively The” – the period is missing at the end of the phrase; “All columns were preconditioned acetonitrile (3 mL) and deionized water (3 mL).” – with preposition is missing; “After eddy current for 1 min” – the verb is missing; “the interaction force between template mole and functional monomers” – after Figure 3;

·         The following phrase should be rephrased, since the supernatant can not be adsorbed: “Urine (10 mL) was measured, and after high-speed centrifugation, 5 mL of supernatant was added to the above solution, and the supernatant was adsorbed by shaking for 10 h.

·         What was the rationale for applying the eddy current to urine samples?

·         Figures 3 and 4 present the “absorption spectra” and not the “adsorption spectra”.

·         The caption of figure 10 is missing the letter “(a)” corresponding to the CAP standard solution analysis

·         In figure 11, standard deviation error bars should appear. The analyses should be realized in triplicate and standard deviation should be determined.

·         English should be modified by a native speaker

In conclusion, this manuscript is suitable for publication, but not in the current state because of the poor English and needs a major revision.

Author Response

Dear Reviewers:

Thank you very much for your letters and comments concerning our manuscript entitled “Preparation of Molecularly Imprinted Polymer Microspheres for Selective Solid-phase Extracting of Capecitabine in Urine Samples.” (Revised title) (Polymers-1896718). The comments are all valuable and very helpful for revising and improving our paper. The manuscript has been revised after consideration of all suggestions and substantial corrections have been made to the revised manuscript which we hope now meets your approval. The main corrections in the paper and responses to the referees’ comments are as follows:

Question 1

The first paragraph of the introduction should be rephrased and discussed about the challenges which occur in analyzing biological samples and not food samples.

Response:

In the preface of the paper, the challenges of biological sample analyses were added and the corresponding references have been modified in the revised manuscript.

Question 2

“MISPE technology is proposed to separate capecitabine (CAP) and its structural analogs in urine samples” – in this study the structural analogues of CAP are not enriched using the developed MIP, only the template molecules. There is no difference between the MIP and NIP behavior towards CYT.

Response:

Thank you very much for this observation. Our research group added CAP and its structural analog CYT to human urine and used MIPs and NIPs to separate and enrich the target molecules in urine samples. MIPs demonstrated strong selective recognition performance for CAP, and the separation of CPT was only through a non-specific recognition process. The results were similar for the recognition process of CAP and CYT by NIPs, mainly due to the lack of matching CPA in the NIPs. Therefore, there is no selective performance for the recognition of CAP and CYT by NIPs.

Question 3

There are some typographical mistakes throughout the manuscript: “polymerizatio,” – in the last paragraph of the introduction, “39.8%, respectively The” – the period is missing at the end of the phrase; “All columns were preconditioned acetonitrile (3 mL) and deionized water (3 mL).” – with preposition is missing; “After eddy current for 1 min” – the verb is missing; “the interaction force between template mole and functional monomers” – after Figure 3;

 (a) Question

 “polymerizatio,” – in the last paragraph of the introduction, “39.8%, respectively The” – the period is missing at the end of the phrase;

Response:

“polymerization” is the correct word form and the added period has been modified in the revised manuscript.

 (b) Question

 “All columns were preconditioned acetonitrile (3 mL) and deionized water (3 mL).”

Response:

The correct sentence “All columns were preconditioned with acetonitrile (3 mL) and deionized water (3 mL), respectively.” has been modified in the revised manuscript.

(c) Question:

“the interaction force between template mole and functional monomers” – after Figure 3

Response:

The sentence "UV spectroscopy has been used to study the interaction force between template mole and functional monomers." has been modified to "UV spectroscopy has been used to study the interaction force between template molecules and functional monomers."

(3) Question:

The following phrase should be rephrased, since the supernatant can not be adsorbed: “Urine (10 mL) was measured, and after high-speed centrifugation, 5 mL of supernatant was added to the above solution, and the supernatant was adsorbed by shaking for 10 h.”

Response:

The correct experiment is described as follows: Subsequently, 10 mL of frozen urine was added to the conical flask de-scribed above, shaken and adsorbed for 10 h. After high-speed centrifugation, the supernatant was taken as a urine sample for testing.

(4) Question:

What was the rationale for applying the eddy current to urine samples?

Response:

Eddy current chromatography separation technology was applied to remove macromolecular substances (including proteins) from the urine samples to avoid negatively affecting HPLC detection.

(5) Question:

Figures 3 and 4 present the “absorption spectra” and not the “adsorption spectra”.

Response:

In Figures 3 and 4, the correct form of “absorption spectra” was added in revised text.

(6) Question:

The caption of figure 10 is missing the letter “(a)” corresponding to the CAP standard solution analysis.

Response:

 Proper annotation (a) of the CAP standard solution in Figure 10 has been added to the revised text.

(7) Question:

In figure 11, standard deviation error bars should appear. The analyses should be realized in triplicate and standard deviation should be determined.

Response:

In Figure 11, all reported data was the averaged result of triplicate tests. The corresponding error bar has been provided in the revised text.

(8) Question:

English should be modified by a native speaker.

In conclusion, this manuscript is suitable for publication, but not in the current state because of the poor English and needs a major revision.

Response:

​The entire paper has been polished with professional structure, and the traces of modification can be seen through the review mode in the revised manuscript.

Thank you and best regards to you!

Yours sincerely,

Renyuan Song, Jiawei Xie, Xiaofeng Yu, Jinlong Ge, Muxin Liu and Liping Guo

yxf@bbc.edu.cn (X.Y.) and songrenyuan0726@163.com (R.S.)

Reviewer 2 Report

This manuscript reports the determination of Capecitabine in Urine Samples Based on Mo-lecularly Imprinted Solid-Phase Extraction. In my opinion, this manuscript can be accepted with minor revisions at Polymers and I would like to address following suggestions to the authors:

1.      The title of the manuscript should be improved to better describe the essence of the work.

2.      The "literature review" section of the manuscript must be improved and is indicated to be from recent years. It is necessary to compare the results of the present study with previous similar studies.

3.      The authors could explain experimental methodologies to validation parameters presented in abstract and to do more information about UV-Vis and HPLC methods in Materials and Methods section.

4.      At Figures 5, 6, 7 and 10, Y-axis title should be added and indicated.

5.      The Conclusions section needs to be rewritten in more clear way to be understandable for the readers. Please, add conclusions about NIPs. MIPs and NIPs microspheres are characterized by FT-IR, SEM, BET and finally is used for pre-concentration of CAP and CYT from urine samples spiked but in conclusions is presented information only by MIPs.

Author Response

Dear Reviewers:

Thank you very much for your letters and comments concerning our manuscript entitled “Preparation of Molecularly Imprinted Polymer Microspheres for Selective Solid-phase Extracting of Capecitabine in Urine Samples.” (Revised title) (Polymers-1896718). The comments are all valuable and very helpful for revising and improving our paper. The manuscript has been revised after consideration of all suggestions and substantial corrections have been made to the revised manuscript which we hope now meets your approval. The main corrections in the paper and responses to the referees’ comments are as follows:

This manuscript reports the determination of capecitabine in urine samples based on molecularly imprinted solid-phase extraction. In my opinion, this manuscript can be accepted with minor revisions at Polymers and I would like to address following suggestions to the authors:

  1. Question:

The title of the manuscript should be improved to better describe the essence of the work.

Response:

According to the manuscript contents, the original title “Determination of Capecitabine in Urine Samples Based on Molecularly Imprinted Solid-Phase Extraction” was modified to “Molecularly imprinted solid-phase extraction of capecitabine from urine samples” in the revised text.

  1. Question: The "literature review" section of the manuscript must be improved and is indicated to be from recent years. It is necessary to compare the results of the present study with previous similar studies.

Response:

In the introduction of the revised text, the analysis methods for the template molecule in biological samples were reviewed along with the latest developments and challenges, while more recent references are now provided.

  1. Question:

The authors could explain experimental methodologies to validation parameters presented in abstract and to do more in-formation about UV-Vis and HPLC methods in Materials and Methods section.

Response:

UV spectroscopy methods to study the interactions between functional monomers and template molecules are provided in the materials and methods section, as detailed in the revised text. More importantly, the relevant content in the abstract was optimized according to the reviewer’s comment.

  1. Question: At Figures 5, 6, 7 and 10, Y-axis title should be added and indicated.

Response:

The ordinate diagrams in Figures 5, 7, and 9 have been added, while the specific ordinate values were not added because the ordinate values have little effect on the results. Figure 6 shows a 1H NMR spectrum and generally there is no need to provide an ordinate prompt when making a map.

  1. Question:

The Conclusions section needs to be rewritten in more clear way to be understandable for the readers. Please, add conclusions about NIPs. MIPs and NIPs microspheres are characterized by FT-IR, SEM, BET and finally is used for pre-concentration of CAP and CYT from urine samples spiked but in conclusions is presented information only by MIPs.

Response:

In the modified manuscript, the conclusions have been rewritten, especially concerning the description of interaction forces between the functional monomer and template molecule using UV spectroscopy. In addition, the characterization of NIPs was also discussed.

Thank you and best regards to you!

Yours sincerely,

Renyuan Song, Jiawei Xie, Xiaofeng Yu, Jinlong Ge, Muxin Liu and Liping Guo

yxf@bbc.edu.cn (X.Y.) and songrenyuan0726@163.com (R.S.)

Round 2

Reviewer 1 Report

The manuscript can be published in the current form.